# Use of Artificial Intelligence to Manage Patient Flow in Emergency Department during the COVID-19 Pandemic: A Prospective, Single-Center Study

**DOI:** 10.3390/ijerph19159667

**Published:** 2022-08-05

**Authors:** Emilien Arnaud, Mahmoud Elbattah, Christine Ammirati, Gilles Dequen, Daniel Aiham Ghazali

**Affiliations:** 1Department of Emergency Medicine, Amiens Picardy University Hospital, 80000 Amiens, France; 2Laboratoire Modélisation, Information, Systèmes (MIS), University of Picardie Jules Verne, 80080 Amiens, France; 3Faculty of Environment and Technology, University of the West of England, Bristol BS16 1QY, UK; 4Amiens Picardy University Hospital—SimuSanté, 80000 Amiens, France; 5INSERM UMR1137, Infection, Antimicrobials, Modelling, Evolution, University of Paris-Diderot, 75018 Paris, France

**Keywords:** COVID-19, artificial intelligence, triage, management of organizations, emergency department

## Abstract

Background: During the coronavirus disease 2019 (COVID-19) pandemic, calculation of the number of emergency department (ED) beds required for patients with vs. without suspected COVID-19 represented a real public health problem. In France, Amiens Picardy University Hospital (APUH) developed an Artificial Intelligence (AI) project called “Prediction of the Patient Pathway in the Emergency Department” (3P-U) to predict patient outcomes. Materials: Using the 3P-U model, we performed a prospective, single-center study of patients attending APUH’s ED in 2020 and 2021. The objective was to determine the minimum and maximum numbers of beds required in real-time, according to the 3P-U model. Results A total of 105,457 patients were included. The area under the receiver operating characteristic curve (AUROC) for the 3P-U was 0.82 for all of the patients and 0.90 for the unambiguous cases. Specifically, 38,353 (36.4%) patients were flagged as “likely to be discharged”, 18,815 (17.8%) were flagged as “likely to be admitted”, and 48,297 (45.8%) patients could not be flagged. Based on the predicted minimum number of beds (for unambiguous cases only) and the maximum number of beds (all patients), the hospital management coordinated the conversion of wards into dedicated COVID-19 units. Discussion and conclusions: The 3P-U model’s AUROC is in the middle of range reported in the literature for similar classifiers. By considering the range of required bed numbers, the waste of resources (e.g., time and beds) could be reduced. The study concludes that the application of AI could help considerably improve the management of hospital resources during global pandemics, such as COVID-19.

## 1. Introduction

Infection with the severe acute respiratory syndrome coronavirus 2 (SARS-CoV-2) can lead to a respiratory syndrome known as coronavirus disease 2019 (COVID-19). This disease first appeared in the Hubei province of China in December 2019, and then rapidly spread across the world [1]. SARS-CoV-2 arrived in Europe via Italy, Spain, and then France (first in the country’s Picardy region, which is home to 1.9 million inhabitants). Amiens Picardy University Hospital (APUH, Amiens, France) was in the frontline, and the initial response to the epidemic was coordinated by the hospital’s emergency medical service (EMS). Thus, APUH had to change its organizational structure in response to this new situation. Firstly, we created a dedicated EMS team to answer the calls about COVID-19 and to triage the suspected COVID-19 patients. The team (comprising medical students and emergency physicians) advised on whether the patient could stay at home or whether an ambulance or a mobile intensive care unit (ICU) should be dispatched [2,3]. This organizational structure enabled the EMS to continue to respond to other medical emergencies. Secondly, we organized a regional bed management system [4] in which an ICU dispatcher managed the intensive care beds and an infectious disease specialist managed the non-intensive-care beds. Thirdly, we changed the organizational structure of the emergency department (ED) as a function of the number of COVID-19 patients, by dedicating a part of the ED to the patients with suspected COVID-19 (with a dedicated care team, reception room, and care areas), replacing it with a short-stay unit for suspected COVID-19 patients when the epidemic intensified, and finally, moving back to a dedicated ED zone. 

Overcrowding in EDs is a concern worldwide. Several studies from different countries have reached the same conclusion: EDs are chronically overcrowded, which leads to unintended breaches and failings in terms of patient safety and the quality of care [5]. In turn, these problems influence patient management. A patient admitted to an inappropriate hospital unit will have a longer length of stay (LOS) and higher care costs [6]. Other issues identified in the literature include a longer hospital stay [6], a higher morbidity rate [7], a higher mortality rate [8,9,10,11], and a greater proportion of patients who leave the ED without being seen [12]. Many of the research groups are working on ways of tackling these problems and limiting the effects of overcrowding: pre-hospital dispatch before attending at the ED (to encourage alternative healthcare provision); better bed coordination; triage by emergency medical staff; front-loading investigations; triage systems; fast tracks; optimized transfer to the destination ward (even if the bed is not ready); and a greater number of available beds [13,14,15].

Determining the number of beds required, respectively, for the COVID-19 patients and non-COVID-19 patients was a real challenge throughout the epidemic. Depending on the demand for the COVID-19 beds, the wards had to be converted from their usual specialty into COVID-19 units. Thus, the remaining COVID-free wards became multidisciplinary, and admitted patients from several specialties [16]. In parallel, we had already begun to work on an Artificial Intelligence (AI) project called “Prediction of the Patient Pathway in the Emergency Department” (3P-U). This project led to the development of a set of software for extracting and preprocessing data, and eventually developing and deploying a predictive model. It consists of a multilayer perceptron that operates on the structured and unstructured data stored in electronic health records (EHRs), and predicts a patient’s hospital admission or discharge at the end of care in the ED [17]. The categorical data are one-hot encoded. The dataset is split randomly, using the conventional ratios of 80% for training and 20% for validation. We have used 3P-U on a daily basis since January 2019, including during the COVID-19 pandemic. The primary objective of the present study was to determine the required number of beds in real time. The second objective was to assess the 3P-U model’s contribution to the fast-track triage versus the standard triage of suspected COVID-19 patients and, presumably, non-COVID-19 patients.

## 2. Materials and Methods

### 2.1. Study Design

We conducted a prospective, single-center study of the patients attending the ED at APUH, in order to validate the 3P-U model’s performance in the specific context of the 2020–2021 epidemic of COVID-19. The present report complies with the Transparent Reporting of a Multivariable Prediction Model for Individual Prognosis or Diagnosis statement [18]. The study was authorized by a hospital committee with for research not requiring approval by an institutional review board, under the reference PI2019_843_0066. The APUH has 1700 beds, a total of 139K patient admissions per year, and 60K ED admissions per year. The ED has three zones: a fast-track zone; the standard ED zone; and a resuscitation room. During the COVID-19 epidemic, we duplicated the fast and the usual ED pathway in the dedicated COVID-19 zone, and we dedicated two beds in the resuscitation room to suspected COVID-19 patients.

### 2.2. Participants

We included all of the adults attending the ED (downstream of the nursing triage stage) from 1 December 2019 to 31 December 2021. We excluded the patients who were immediately transferred to the resuscitation room upon arrival and those with no triage data. We used the 3P-U predictive model, as deployed at APUH. The 3P-U model predicted the physician’s final decision (admission vs. discharge) on the basis of the triage data. The model had been trained on the data from 233,814 patients who attended the ED between 2015 and 2018. In an internal validation and with a threshold of 0.5, the area under the receiver operating characteristic curve (AUROC) was 83%, the precision (i.e., the positive predictive value) was 72%, and the recall (i.e., the sensitivity) was 62% [17]. The predictors were: the demographic characteristics (age and sex); the clinical triage characteristics (heart rate, blood pressure, blood oxygen saturation, body temperature, capillary blood glucose level, capillary blood ketone level, oxygen flow, hemoglobin capillary test, expired breath alcohol level, pain intensity, urine tests, and the French Emergency Nurses Classification in Hospital (FRENCH) triage scale [19]); the non-clinical triage characteristics (the arrival or referral route (referred by a physician, an accident in the workplace, a sports accident, etc.), whether or not the patient was accompanied, the waiting status (sitting, stretchered, etc.), the family context, the time of arrival); and the unstructured data (triage notes and the medical history). In the specific context of the ED, some of the data were not collected on purpose (DNPC) and do not correspond to the missing data per se (for example, the capillary blood glucose level was not measured for the patients with straightforward trauma injuries). The method for completing the missing values depended on the type of variable (see the Appendix A).

### 2.3. Intervention

On arrival at the ED, the nurses categorized the patients according to the FRENCH triage scale and flagged the patient as probably having COVID-19 or not. All of the data reported in the hospital’s electronic health records (Resurgences^®^, Berger-Levrault, Boulogne-Billancourt, France) were automatically extracted by 3P-U, and then preprocessed using the same pipeline as in the development stage. The prediction was calculated and stored in the database. Using a nasopharyngeal swab, the suspected COVID-19 patients were tested with a lateral flow test (Panbio COVID-19 Ag (Nasal Version), Abbott Chicago, IL, USA), and then a confirmatory PCR test (TaqPath COVID 19 CE-IVD RT-PCR Kit, ThermoFisher, Waltham, MA, USA). Neither the flag for suspected COVID-19, the lateral flow test result, nor the PCR test result were included as features in the model, since these variables were absent from the training phase. The bed management team had access to a dashboard with the COVID-19 patients’ real-time status: “not known”; “to be admitted” (decided by a physician); “likely to be admitted” (3P-U’s recommendation); “likely to be discharged” (3P-U’s recommendation); or “to be discharged” (decided by a physician). The development stage and the validation stage differed with regard to the context (before vs. during the COVID-19 epidemic) and the interpretation of the probability of admission. We defined “unambiguous patients” as those for whom the predicted probability of admission was below 0.2 (interpreted as “likely to be discharged”) or over 0.6 (interpreted as “likely to be admitted). Between 0.2 and 0.6, we considered that the patient was in a “grey zone” and was flagged as “status not known”.

### 2.4. Statistical Analysis

The continuous variables were described as the mean (standard deviation (SD)), and the categorical variables were described as the number of patients (percentage). To describe the population, we retrieved the PCR test results performed in the ED. The COVID-19 PCR tests were performed on all of the admitted patients and the symptomatic patients before admission. The model’s performance was assessed with regard to the AUROC. We modelled two sets of patients: all of the patients (to determine the maximum number of beds required); and “unambiguous patients” (to assess the minimum number of beds required). The 3P-U’s decision and the final decision by a physician were compared based on precision, recall, and accuracy for the two sets of patients. Precision was defined as the ratio between the true detected admissions and all of the detected admissions. The recall was defined as the ratio between the true detected admissions and all of the admissions. The accuracy was defined as the proportion of the patients with a correct classification.

## 3. Results

### 3.1. Characteristics of the Study Population

Between 1 December 2019 and 31 December 2021, 136,139 patients attended the APUH ED. After the exclusion of 30,682 patients (4044 immediately admitted to the ICU, 21,991 with missing triage data, and 4647 who left without being seen), we included 105,457 patients. According to the physician’s final decision, a total of 33,841 (32%) patients were admitted (21,470 (21%) were admitted to a medical ward; 7604 (7%) were admitted to a surgical ward; and 4641 (4%) were admitted to the ICU) and 71,616 (68%) were discharged. The FRENCH triage scale ratings and other patient characteristics are summarized in Table 1. Figure 1 shows the changes over time in the PCR results in the Somme county of France (where the APUH is located) during the inclusion period. The complete dataset and the DNPC are described in Appendix A.

### 3.2. The Model’s Performance

Regarding the classification accuracy, the model’s AUROC was 0.82 for all of the patients and 0.90 for the unambiguous patients (Appendix B). According to the interpretation rules described above, 38,353 (36.4%) patients were flagged by 3P-U as “likely to be discharged”; 18,815 (17.8%) patients were flagged as “likely to be admitted”; and 48,297 (45.8%) patients could not be flagged. With regard to the accuracy (defined as the number of correctly classified patients as a proportion of the total number of flagged patients), 49,377 (86.4%) of the 57,168 flagged patients were correctly classified. Of these 49,377 patients, 13,840 (24.2%) were correctly classified as “likely to be admitted”, and 35,537 (62.2%) were correctly classified as “likely to be discharged”. In contrast, 7791 (13.6%) patients were wrongly classified: 4975 (8.7%) patients were wrongly classified as “likely to be admitted” but were finally discharged; and 2816 (4.9%) patients were misclassified as “likely to be discharged” but were finally admitted. Hence, for the unambiguous patients, the recall (i.e., sensitivity) was 83.1%, the precision was 73.6%, the specificity was 87.7% and the predictive negative value was 92.7% (Figure 2). Based on the minimum number of admissions (for the unambiguous patients only) and the maximum number of admissions (for all of the patients) (Figure 3), the bed managers coordinated the conversion of wards into dedicated COVID-19 units. The bed managers also used this indicator to roll back the dedicated COVID-19 units and re-establish the ward’s initial specialty. The 3P-U dashboard was considered more efficient than manual estimation.

### 3.3. Individual Predictions

At the triage stage, the nurses classified 4031 (3.8%) patients as probably having COVID-19 and 101,426 (96.2%) as probably not having non-COVID-19. In the COVID-19 group, 3P-U flagged 768 (0.7%) probable admissions and 1420 (1.3%) probable discharges. In the non-COVID-19 group, 3P-U flagged 18,018 (17.1%) probable admissions and 36,886 (35.1%) probable discharges (Figure 4).

## 4. Discussion

### 4.1. The 3P-U threshold

The AUROC for the 3P-U model was 82%, which is in the middle of the range of values reported in the literature for similar classifiers [21]. The choice of the threshold was difficult to address. In our earlier work [6], we chose 0.5 as the threshold because it gave the highest F1-score (an indicator based on the harmonic mean of the precision and the recall): in the present study, the precision was 72% and the recall was 62% (Appendix D). The presence of the false positives and false negatives meant that time and resources were wasted: bed managers booked beds for patients who were finally discharged. Hence, to minimize the false positives and false negatives, we next considered the unambiguous patients only (i.e., those with admission probabilities at each end of the scale). If the probability of admission is below 0.2, the patient will almost certainly be discharged. Conversely, if the probability of admission is over 0.6, the patient will almost certainly be admitted. When the model was applied to the unambiguous patients, the AUROC was 0.90, which corresponds to very good performance. Not interpreting the predictions for ambiguous cases is a safety strategy for the patients and is profitable for the caregivers (i.e., time savings).

### 4.2. Related Work

Researchers have developed several approaches for trying to reduce ED congestion using machine learning, e.g., enhancing triage [22] or predicting a diagnosis (such as stroke [23] or ischemic heart disease [24]). Graham et al.’s model [25] used the same features as the 3P-U model to predict the same outcome. The AUROC for Graham et al.’s model (0.86) was higher than that of the 3P-U. Although Graham et al. did not specify how they selected the threshold, their model had a sensitivity of 54% and a specificity of 90%. We selected a different threshold, and so the 3P-U was more sensitive (0.597, vs. 0.537 for Graham et al.’s model) but less specific (0.650, vs. 0.903 for Graham et al.’s model). Unfortunately, Graham et al. did not report their precision, so the models’ respective performances have to be compared with regard to specificity. Our present results highlight the importance of threshold selection: the AUROC reflects the model’s overall performance, whereas sensitivity and specificity reflect the performance for a real decision. Hong et al. [5] trained their model (using similar features) to predict admission. Their highest AUROC was 92% (using all of the features), with a sensitivity of 82% and a specificity of 85%. Hong et al. chose their threshold and compared their models by setting the specificity to 85%. Tan et al. [26] built a model that predicted the patient’s outcome on the basis of the triage data, the medical history, and the laboratory data. The highest AUROC for admission prediction was 84%, using a random forest model. Tan et al. incorporated the model into a web service and added the latter to the hospital’s information system for operational use. Many researchers [5,25] have built models in a laboratory environment, but few have reported on operational use.

### 4.3. Implications

When we started to apply the model in production, we observed that the “grey zone” patients were not always correctly categorized; this meant that resources were wasted. Indeed, the interpretation of the prediction given to the bed manager caused more confusion than we expected. For example, the manager started to look for a bed, but the patient was finally discharged. As a result, we considered all of the patients (to calculate the maximum number of beds required) and the unambiguous patients only (to calculate the minimum number of beds required). The bed manager received this information in real time, which helped him/her to optimize the search for beds. To prevent full occupancy of the specialist beds in APUH’s wards, the bed manager had to send the patients not requiring specialist care to smaller medical centers. This search was time-consuming, and so we developed a specific application for flagging up free beds [4]. This overview of the numbers of beds required for the suspected COVID-19 patients and the presumed non-COVID-19 patients was clearly of value to the bed manager, and the patient-by-patient prediction helped to attribute beds.

Although the 3P-U saved the bed manager time, the final decision on the patient’s destination was taken by the ED’s physicians. In their review, Sanchez-Salmeron et al. [22] stated that all of the automatic medical predictions must be checked by a human. This is particularly important in novel situations (such as the COVID-19 epidemic), of which physicians have little experience. The possibility of classification errors must always be considered. In the present study, the risk was limited by the physician’s final decision. The 3P-U helped to organize the ED. Each day, a physician acted as a “flow manager”. She/he had an overview of the attending patients, the level of congestion in each sector, and possible bottlenecks. Developing a manual summary is time-consuming, and the summary might not be up to date when the ED is overcrowded. The 3P-U automatically provides this overview, along with a dashboard (Appendix C). Although the very urgent cases will always be obvious for the physicians, 3P-U’s algorithm can classify patients more rapidly, and for many patients in parallel. This is likely to give physicians more time to care for patients.

The APUH is a regional referral center: no patients are diverted to other hospitals, and patients are sometimes diverted from smaller regional hospitals to APUH. The increase in patient flow during the COVID-19 epidemic did not change this strategy, regardless of 3P-U’s deployment.

### 4.4. Strengths and Limitations

Our study claims to have the following set of strengths. First, the 3P-U was validated in real time, using the EHRs, with no additional tasks for the triage nurse. Secondly, the application was deployed almost in the same way as in production and was tested in real-life scenarios. However, it is acknowledged that the 3P-U model’s performance was not the same as under experimental conditions: it was trained on the data from the patients attending the ED before the COVID-19 epidemic, who differed from those attending the ED during the epidemic [27]. Despite this difference, the model’s performance (AUROC = 82%) was generally acceptable, and the model could be used to anticipate bed attributions. Thirdly, we made the prediction safer by applying safety margins: the probabilities between 0.2 and 0.6 were not interpreted.

The present study also has some limitations. First, this was a single-center study, and hence, the model was trained with the data from one hospital only. It is conceived that the model could be conveniently trained in another center, but it would not be transferrable “as is”. Indeed, our model is representative of the APUH’s population. Using the current model in another medical center without a specific training stage would reproduce the strategies and procedures used in the APUH, which might not be applicable. Secondly, we only considered the triage data, which contains less medical information than the data obtained later in the care pathway. Thirdly, we considered unambiguous patients only, rather than the full population. Fourthly, we did not study the direct impact on the patient. The LOS and admission to the appropriate ward would probably be good indicators of this impact. However, the changes in the organization and the greater number of patients admitted during the COVID-19 epidemic would probably have biased the measurements of the LOS.

## 5. Conclusions

The 3P-U AI model helped to manage patients in the APUH ED during the COVID-19 epidemic by predicting the maximum and minimum numbers of beds required. It also helped to categorize the suspected COVID-19 patients and the presumed non-COVID-19 patients in the fast-track and standard ED pathways. The model was utilized effectively for categorizing the individual patients and providing an overview of predictions (the total size of the various categories of patient). In view of that, the present study confirms that AI models can provide a vital support for streamlining the ED operations, and also for curbing the waste of resources. Future studies should focus on the two remaining problems: predicting the definitive diagnosis and predicting the patient’s LOS in a ward after transfer from the ED. The admission to the right specialty ward is known to be associated with a lower mortality rate, and prediction of the LOS would help to optimize the use of hospital resources.

## Figures and Tables

**Figure 1 ijerph-19-09667-f001:**
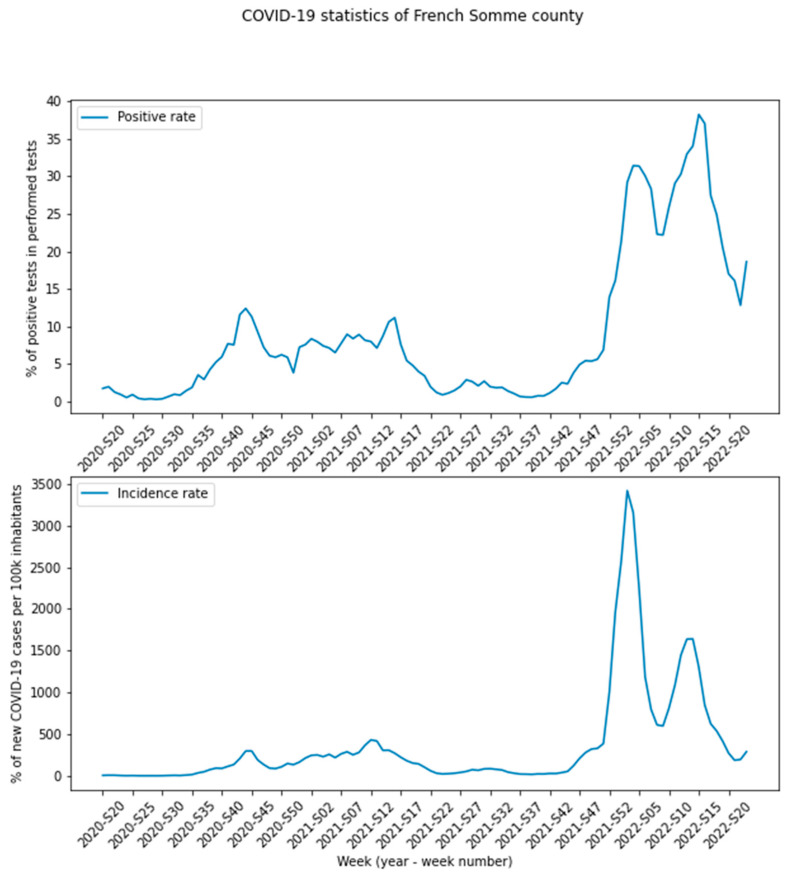
Changes over time in the PCR results in the Somme county of France. The data were extracted from data.gouv.fr, accessed on 1 July 2022 [20].

**Figure 2 ijerph-19-09667-f002:**
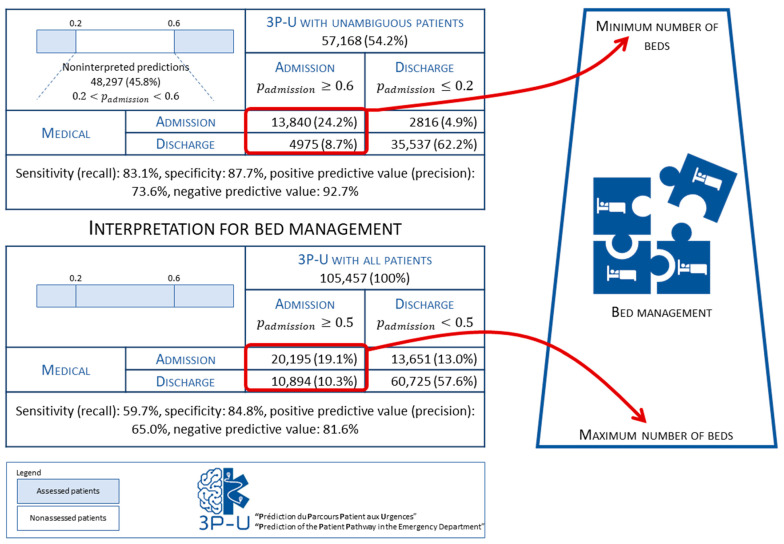
The 3P-U model’s performance and the bed manager’s interpretation.

**Figure 3 ijerph-19-09667-f003:**
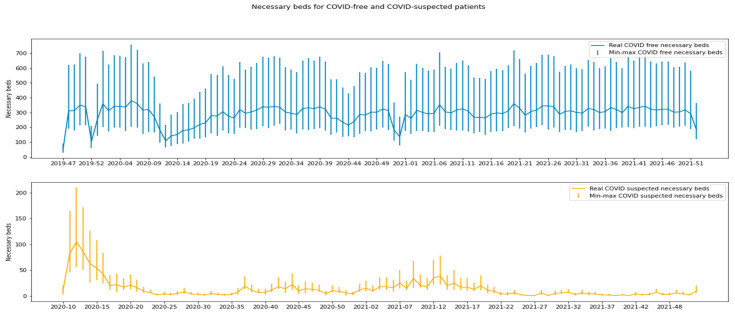
Beds required for presumed non-COVID-19 patients and suspected COVID-19 patients by week, as used by the bed manager to optimize organization of the wards.

**Figure 4 ijerph-19-09667-f004:**
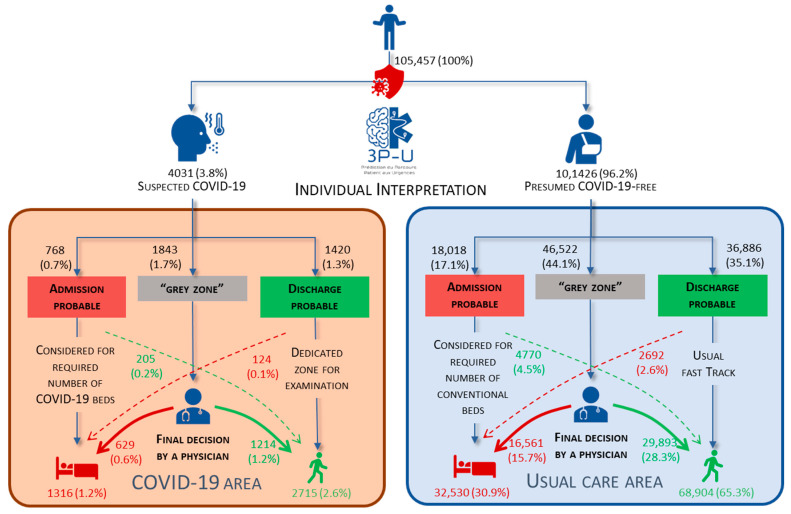
Individual triage prediction in the Amiens Picardy University Hospital (Amiens, France) ED.

**Table 1 ijerph-19-09667-t001:** Characteristics of the study population.

Demographic Characteristics	Overall
Number of patients, *n* (%)	105,457 (100%)
Age, mean ± SD	51 ± 22
Sex, *n* (%)	
Male	50,639 (48%)
Female	54,818 (52%)
**Clinical triage characteristics**	
Heart rate (/min), mean ± SD	86 ± 18
Systolic blood pressure (mmHg), mean ± SD	138 ± 24
Diastolic blood pressure (mmHg), mean ± SD	79 ± 23
Blood oxygen saturation (%), mean ± SD	99 ± 2
Body temperature (°C), mean ± SD	36.4 ± 0.7
Capillary blood glucose level (mmol/L), mean ± SD	7.33 ± 4.19
Capillary blood ketone level (mmol/L), mean ± SD	0.98 ± 1.97
Oxygen flow (L/min), mean ± SD	0.6 ± 3.7
Capillary blood hemoglobin level (dg/dL), mean ± SD	11.72 ± 2.96
Expired breath alcohol level (g/L), mean ± SD	1.81 ± 0.78
Bladder volume (mL), mean ± SD	334 ± 305
Pain intensity, mean ± SD	3 ± 3
Patient rating on the FRENCH triage scale, *n* (%)	
1	235 (< 1%)
2	3975 (4%)
3	56,679 (54%)
4	28,363 (27%)
5	14,849 (14%)
**Outcome**	
Admission to a medical ward	21,470 (21%)
Admission to a surgical ward	7604 (7%)
Admission to the ICU	4641 (4%)
Discharge	71,616 (68%)

## Data Availability

Data available on request due to restrictions related to privacy regulations.

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
