# Peer review of "Use of Artificial Intelligence to Manage Patient Flow in Emergency Department during the COVID-19 Pandemic: A Prospective, Single-Center Study"

_ijerph, 2022, doi:10.3390/ijerph19159667_

Round 1

Reviewer 1 Report

1. In other studies, are there other machine learning or deep learning approaches that could help reduce emergency department congestion?

2. Is it appropriate to set the threshold to 0.5 for the previously trained 3P-U model? Are there other thresholds for larger AUROCs?

3. Does the 3P-U model have an impact on fast diversion and standard diversion? I don't see it in the results.

Author Response

Dear Editor-in-Chief, dear Guest Editor,

Thank for giving us an opportunity to submit a revised manuscript. We also thank the reviewers for their comments, which have markedly improved the quality of the manuscript. Please find below our point-by-point answers.

Please do not hesitate to contact me if any aspects require further explanation.

Kind regards,

Reviewer 1

1. In other studies, are there other machine learning or deep learning approaches that could help reduce emergency department congestion?

Thank you for highlighting this point.  We have included references to other similar projects in section 4.2 “Related work”.

2. Is it appropriate to set the threshold to 0.5 for the previously trained 3P-U model? Are there other thresholds for larger AUROCs?

Thank you for pointing out that this aspect is not enough clear in the manuscript. The threshold is independent of the AUROC and is one point on the curve. In our model, a threshold of 0.5 gave the best F1 score. The F1 score is the harmonic mean of the precision and the recall. We defined the F1 score in Appendix D.

3. Does the 3P-U model have an impact on fast diversion and standard diversion? I don't see it in the results.

Thank you for this relevant question. Regardless of whether or not 3P-U is used, patients are never diverted to other hospitals because Amiens Picardy University Hospital is the regional referral center. The increase in patient flow during the COVID-19 period did not change this strategy; (we now mention this point in the Discussion). Although 3P-U probably had an impact on the length of stay (LOS), we did not study this variable; in the context of COVID-19, the changes in organization and patient flow biased the LOS during the study. We now mention this point as a (fourth) study limitation.

Reviewer 2 Report

The paper “Use of Artificial Intelligence to Manage Patient Flow in Emergency Department During the Covid-19 pandemic: A Prospective, Single-Center Study” present efficient application of AI in medicine. Topic of the manuscript suits well to International Journal of Environmental Research and Public Health and sourly will be interesting to readers. In my opinion manuscript can be published after minor revision according to listed comments:

Line 87. Please include some more information about 3P-U model (if possible)? What type of artificial intelligence model was included into model?

Line 149. Please improve

Table 1. I suggest presenting the results as an average ± SD

Line 164. Please improve

Line 188. Please improve

Author Response

Dear Editor-in-Chief, dear Guest Editor,

Thank for giving us an opportunity to submit a revised manuscript. We also thank the reviewers for their comments, which have markedly improved the quality of the manuscript. Please find below our point-by-point answers.

Please do not hesitate to contact me if any aspects require further explanation.

Kind regards,

Reviewer 2

The paper “Use of Artificial Intelligence to Manage Patient Flow in Emergency Department During the Covid-19 pandemic: A Prospective, Single-Center Study” present efficient application of AI in medicine. Topic of the manuscript suits well to International Journal of Environmental Research and Public Health and sourly will be interesting to readers. In my opinion manuscript can be published after minor revision according to listed comments:

We thank the reviewer for his/her comments.

Line 87. Please include some more information about 3P-U model (if possible)? What type of artificial intelligence model was included into model?

The 3P-U model has been described in the IEEE conference paper cited in the manuscript. Nevertheless, to be more comprehensive, we now mention some of the model’s characteristics.

Arnaud E, Elbattah M, Gignon M, Dequen G. Deep Learning to Predict Hospitalization at Triage: Integration of Structured Data and Unstructured Text [Internet]. In: 2020 IEEE International Conference on Big Data (Big Data). Atlanta, GA, USA: IEEE; 2020 [cité 2021 mai 18]. page 4836‑41.Available from: https://ieeexplore.ieee.org/document/9378073/

Line 149. Please improve

Thank you for your comment. The reference has been updated.

Table 1. I suggest presenting the results as an average ± SD

Thank you for this remark. We have changed the format of Table 1 and Appendix A.

Line 164. Please improve

Thank you for highlighting this misunderstanding. We now specify the meaning of “49,377” – the number of patients. We now also define “accuracy” more precisely.

Line 188. Please improve

Thank you for your comment. We have corrected the reference.

Reviewer 3 Report

The authors performed a prospective, single-center study of patients attending APUH’s ED in 2020 and 2021. The main objective was to determine the minimum and maximum numbers of beds required in real-time, according to the 3P-U model. It is a significant practical problem.

 Based on the predicted minimum and a maximum number of beds, the hospital management coordinated the conversion of wards into dedicated COVID-19 units. The study concludes that the application of AI could primarily help improve the management of hospital resources during global pandemics such as COVID-19. The paper should be accepted after improvement. The list of my comments is as follows:

1. The introduction should be extended. There are only six references - please add additional references during the extension of this section.

2. There is one very interesting issue - Artificial Intelligence is observed only one time in the whole work. What AI methods were used? I see only statistical analysis. It should be better explained. Maybe a little change in the title or in the abstract is needed.

3. Do you use in your work only statistical analysis based on the results AI project?

4. On page 4 is missing a reference (Reference source not found).

5. Fig 1 x-axis is no caption.

6. The results can be presented as interval numbers. It has sense because [min number, max number] is an interval that should be allocated the real state.

7. Conclusion should be extended. Please also add future research directions.

8. References should also be extended.

Author Response

Reviewer 3

The authors performed a prospective, single-center study of patients attending APUH’s ED in 2020 and 2021. The main objective was to determine the minimum and maximum numbers of beds required in real-time, according to the 3P-U model. It is a significant practical problem.

Based on the predicted minimum and a maximum number of beds, the hospital management coordinated the conversion of wards into dedicated COVID-19 units. The study concludes that the application of AI could primarily help improve the management of hospital resources during global pandemics such as COVID-19. The paper should be accepted after improvement. The list of my comments is as follows:

  1. The introduction should be extended. There are only six references - please add additional references during the extension of this section.

Thank you for this comment. We added have content to the Introduction and cite appropriate references.

  1. There is one very interesting issue - Artificial Intelligence is observed only one time in the whole work. What AI methods were used? I see only statistical analysis. It should be better explained. Maybe a little change in the title or in the abstract is needed.

Thank you for highlighting this point. This study addresses the application of the 3P-U model in the context of a COVID-19 epidemic. We built the 3P-U model inside an AI model, which consists of a multilayer perceptron. The model has been described in the IEEE conference paper cited in the manuscript, and we now mention some of its characteristics in Introduction.

Arnaud E, Elbattah M, Gignon M, Dequen G. Deep Learning to Predict Hospitalization at Triage: Integration of Structured Data and Unstructured Text [Internet]. In: 2020 IEEE International Conference on Big Data (Big Data). Atlanta, GA, USA: IEEE; 2020 [cité 2021 mai 18]. page 4836‑41.Available from: https://ieeexplore.ieee.org/document/9378073/

  1. Do you use in your work only statistical analysis based on the results AI project?

Thank you for this relevant question. Yes, we apply only a statistical analysis based on the 3P-U prediction and two thresholds to the study population.

  1. On page 4 is missing a reference (Reference source not found).

Thank you, we have corrected the hyperlink.

  1. Fig 1 x-axis is no caption.

Thank you for highlighting this lack of precision. To make this easier to understand, we have applied the same label to the X-axis on both graphs and also define the Y-axis label.

6. The results can be presented as interval numbers. It has sense because [min number, max number] is an interval that should be allocated the real state.

Thank you for your comment. Do you want us to present the data as median [range], rather than mean ± SD? Reviewer 1 has asked us to quote the data as the mean ± SD (previously “mean (SD)”)

7. Conclusion should be extended. Please also add future research directions.

Thank you for this comment. We have extended the conclusion by adding some perspectives.

8. References should also be extended.

Thank you for your comment. We have added several references in response to the various requests. We now cite 27 references.

Round 2

Reviewer 3 Report

The paper has been improved and can be accepted in its current form.